# Soldier Caste-Specific Protein 1 Is Involved in Soldier Differentiation in Termite *Reticulitermes aculabialis*

**DOI:** 10.3390/insects13060502

**Published:** 2022-05-26

**Authors:** Zhiwei Wu, Yunliang Du, Zhenya Li, Ruiyao Guo, Yiying Li, Jizhen Wei, Xinming Yin, Lijuan Su

**Affiliations:** 1College of Life Science, Henan Agricultural University, Zhengzhou 450002, China; mrfish12306@163.com (Z.W.); yunliangdu@163.com (Y.D.); guoruiyao999@163.com (R.G.); ryujje@163.com (Y.L.); 2College of Plant Protection, Henan Agricultural University, Zhengzhou 450002, China; zhenya0371@163.com (Z.L.); weijizhen1986@163.com (J.W.)

**Keywords:** soldier differentiation, soldier caste-specific protein 1, presoldier, molting, termite, juvenile hormone

## Abstract

**Simple Summary:**

Workers can be differentiated into soldiers that have distinctive defensive morphologies using Juvenile hormone (JH) induction, which is transported to target tissues from its site of secretion by high affinity carriers. In this study, *soldier caste-specific protein 1* (*RaSsp1*) in *Reticulitermes aculabialis* was cloned owing to its high expression in soldiers. RaSsp1 was homologous with the JH-binding protein and could combine JHIII by Predict Protein online. *RaSsp1* RNAi exerted a great effect on soldier differentiation, as evidenced by the shorter head capsule, reduced mandible size, delayed molting time and decreased molting rate, as well as downregulated the expression of JH signal related-gene in soldiers. These results suggested that RaSsp1 may be involved in soldier differentiation by binding and transporting JH.

**Abstract:**

Termite soldiers are a unique caste among social insects, and their differentiation can be induced by Juvenile hormone (JH) from workers through two molts (worker–presoldier–soldier). However, the molecular mechanism underlying the worker-to-soldier transformation in termites is poorly understood. To explore the mechanism of soldier differentiation induced by JH, the gene *soldier caste-specific protein 1* (*RaSsp1*, NCBI accession no: MT861054.1) in *R. aculabialis* was cloned, and its function was studied. This gene was highly expressed in the soldier caste, and the protein RsSsp1 was similar to the JHBP (JH-binding protein) domain-containing protein by Predict Protein online. In addition, JHIII could be anchored in the hydrophobic cage of RaSsp1 as the epoxide of the JHBP-bound JH according to the protein ligand molecular docking online tool AutoDock. The functional studies indicated that knocking down of the *RaSsp1* shorted the presoldier’s head capsule, reduced mandible size, delayed molting time and decreased molting rate (from worker to presoldier) at the beginning of worker gut-purging. Furthermore, knocking down of the *RaSsp1* had a more pronounced effect on soldier differentiation (from presoldier to soldier), and manifested in significantly shorter mandibles, rounder head capsules, and lower molting rate (from worker to presoldier) at the beginning of presoldier gut-purging. Correspondingly, the expressions of JH receptor *Methoprene-tolerant* (*Met*), the JH-inducible transcription factor *Krüppel homolog*1 (*Kr-h1*) and ecdysone signal genes *Broad-complex* (*Br-C*) were downregulated when knocking down the *RaSsp1* at the above two stages. All these results that RaSsp1 may be involved in soldier differentiation from workers by binding and transporting JH.

## 1. Introduction

As a group of eusocial insects, termite colonies are composed of various castes with a clear labor division. That is, the workers generally perform foraging, tunneling, and brood tending, and the soldiers only engage in the defense of their colonies with their distinctive defensive morphologies that are not seen in any of the other castes, such as enlarged mandibles, brush-like labrum, and frontal projection (called nasus) [1,2]. Termites are the most phenotypically plastic insects, and the evolution of the unique postembryonic developmental plasticity of termites constitutes the foundation of their social life, which provides many significant biological clues for understanding the phenotypic plasticity mechanism, which is also important for deducing the common mechanisms of sociality in other social insects such as aphids and thrips [2,3,4]. Importantly, exploring the phenotypic plasticity mechanism may contribute to the control of termites by providing some important target genes for RNAi strategies or for designing new insecticides.

Although several reports showed that caste fates of termites are determined by genetic factors [5], some environmental factors such as season and/or social interactions among colony members could affect the termite’s differentiation [6,7]. Likewise, Nijhout and Wheeler [8] proposed a model that varying juvenile hormone (JH) titers at the start, mid-phase, and end of each intermolt period could account for the developmental diversity, and this was supported by many documents on some termite species, such as *Cryptotermes secundus* [9,10], *Hodotermes sjostedti* [11] and *R. aculabialis* [12,13]. Specifically, in some cases of soldier differentiation, the soldier-caste fate could be changed during the period of postembryonic development, when soldier differentiation from workers was induced by juvenile hormone (JH) or JH analogue (JHA) with two molting and a presoldier stage occurring in termites, forming a transformation of “worker–presoldier–soldier” [2,14].

In addition, during soldier differentiation, JH may participate in not only the soldier molting process but also the morphogenesis of soldier special weapons, especially the mandibles [15,16,17,18]. Thus, as the most critical endocrine regulator to control soldier differentiation, the JH signaling pathway have been the focus for exploring the phenotypic plasticity mechanism [3,19]. Meanwhile, several up- and downstream genes in the JH signaling pathway have been studied [2], such as *Met* (Methoprene-tolerant, JH receptor gene) [17,20], *Kr-h1* (Krüppel-homolog 1) [18], *hexamerin*, JH binding protein (*Hex*-1 and *Hex*-2) [21], P450, and tyrosine metabolic pathway genes [22]. For example, in *Zootermopsis nevadensis*, soldier differentiation could be activated by the nuclear receptor of JH Met [17]. The JH signaling pathway (Met-Kr-h1-E93) could work as a regulator of caste differentiation in termites [23]. *Kr-h*1, an early response gene induced by JH, is also involved in insect metamorphosis and reproduction [24,25]. The expression of *Broad-complex* (Br-C) is repressed by JH and is induced by the molting hormone, and the Kr-h1 protein represses the pupal specifier through directly binding to the KBS sequence in the BR-C promoter [26]. *Hex*-1 and *Hex*-2 were reported to participate in the downregulation of JH titer in workers to suppress juvenile-hormone-dependent worker differentiation to the soldier-caste phenotype in *Reticulitermes flavipes* [21]. In addition, relevant research studies have also showed that JH regulates the morphogenesis of soldier-specific weapons through coupling 20E, TGFβ and insulin signaling pathways [2,27,28,29].

As a suitable insect for exploring the phenotypical plasticity, the caste of *R. aculabialis* branches into the sexual nymph line and the worker line, between which the workers can develop in one of the three ways: remain as a worker, transform to a soldier, or become a neotenic reproductive (secondary reproductives) [12,30]. Similar to other termite species, the soldier differentiation of *R. aculabialis* can also be induced by JH, and the induction rate can reach up to 60% [13]. To date, nearly all studies focus on the JH-regulated pathway in termites for studying the phenotypic plasticity mechanism; however, investigations on the proteins that act in the delivery of JH to target tissues in soldier differentiation have rarely been reported. JHBP (JH binding protein) and take-out are hemolymph carrier proteins of Lepidoptera, which can deliver JH to target tissues [31,32]. The binding assays of JH with JHBP suggested that two cysteine residues at the N-terminus form a disulfide bond that is the most important for ligand recognition [33,34]. As for termites, this type of hemolymph carrier protein was named soldier-specific protein (Ntsp1) based on the protein structure and expressed pattern [35]. Nevertheless, the function of Ntsp1 in soldier differentiation has yet to be studied.

In this study, a *RaSsp1* gene was cloned from the transcriptome database of *R. aculabialis* owing to its high expression in the soldier caste (unpublish data). The structure prediction indicated that RaSsp1 was similar to the JHBP (JH-binding protein) domain-containing protein by Predict Protein online, and that JHIII could be anchored in the hydrophobic cage of RaSsp1 as the epoxide of the JHBP-bound JH by 3D structure prediction with AutoDock. The function studies by RNAi at the beginning of worker gut-purging and presoldier gut-purging revealed that the RaSsp1 protein affected soldier differentiation and the expressions of the JH receptor *Met*, the JH-inducible transcription factor *Kr-h1*, and ecdysone signal genes *Br-C*. All these findings suggest that *RaSsp1* may be involved in and affect soldier differentiation from workers by binding and transporting JH.

## 2. Materials and Methods

### 2.1. Insect Collection and JH Treatment

Termite *R. aculabialis* colonies were collected from woods in Shangcheng County, Henan Province, China, in April to October 2019. The colonies were brought to the climate chamber and kept in opaque plastic cases at approximately 24 °C under constant darkness. To induce soldiers, workers were picked from rotten wood and treated at starvation for 24 h. A total of 120 μg JHA (Pyriproxyfen, Sigma Chemical Co., St. Louis, MO, USA) was dissolved in 450 μL acetone, and then the buffer was dipped to 68 mm diameter filter paper, with the filter paper kept at room temperature to allow the acetone to volatilize naturally. Afterward, JHA-contained filter paper was placed in a plastic petri dish with 70 mm diameter. To ensure the gut-purging of workers and presoldiers, 600 μL of brilliant blue solution with a concentration of 0.0125 g/mL was added onto the filter paper in the petri dish, followed by a total of 60 workers placed into each petri dish. Finally, the petri dish was placed in the artificial climate incubator at approximately 24 °C under constant darkness.

### 2.2. Sample Preparing

In order to analyze the expression of *RaSsp1* in different termite castes and soldier tissues, RT-qPCR (real-time quantitative polymerase chain reaction) was performed. The caste samples (the head of worker, presoldier and soldier) in the three nature colonies, the part samples (head, thorax and abdomen without intestine) of the soldiers, and the tissue samples (head cuticle, mandible, antennae, hemolymph, fat body and midgut) of the soldiers were obtained through the anatomy of soldier under Stereoscope Leica 205A (Leica Microsystems, Wetzlar, Germany). In addition, for detecting the expression of *RaSsp1* during soldier differentiation induced by JHA, individuals were sampled at the following time points: 0, 1, 5, 9, and 9.5 days after JHA treatment; 0, 1, 5, 9, 9.25, 9.5, and 9.75 days after the 1st molting; and then 0, 1, 5, and 9 days after the 2nd molting. Among these, samples at day 0 in workers without JHA treatment were used as control, and 0 days in presoldier and soldier were shown within 24 h after molting. Each treatment was repeated 3 times (biological replication, 5 different individuals/per replication), and three RT-qPCR assays (technical replication) were performed using the same cDNA samples of each time point.

### 2.3. cDNA Cloning and Homology Search

Total RNA was extracted from the heads of termites with Trizol (TaKaRa, Dalian, China), and then the purity and concentration of the extracted RNA was confirmed using a NanoVue spectrophotometer (Thermo Fisher, Waltham, MA, USA). Further, the quality of the total RNA was verified by the Agilent 2100 bioanalyzer (Agilent, Santa Clara, CA, USA). cDNA synthesis was carried out using reverse transcription kit (Vazyme, Nanjing, China). PCR was performed with the specific primer (*RaSsp1*-ORF (open reading frame)-F: 5′-ATGCACCTCATTGTCCTTCT-3′, 5′-TCATTCTGTTGATAATCCTG-3′) that was designed according to the result of transcriptome sequencing. The analysis of homology and clustering from the predicted amino acid sequence was aligned via the NCBI database and MEGA7 (7.0.26). The 2D and 3D structures of RaSsp1 were predicted with SignalIP-5.0 (http://www.cbs.dtu.dk/services/SignalP/ accessed on 5 April 2022) and SWISS-MODEL (https://swissmodel.expasy.org/interactive accessed on 6 April 2022). The topology and overall structure of RaSsp1 with JH was created using protein ligand molecular docking online tool AutoDock Vina V.1.2.0 software.

### 2.4. Real-Time Quantitative PCR

The RNA sample of termites was treated with DNase I (Fermentas, Waltham, MA, USA) to remove potential DNA contamination and was reversely transcribed into first-strand cDNA with First-Strand cDNA Synthesis Kit (Fermentas, Waltham, MA, USA). To test the expression levels of *RaSsp1*, *Met*, *Kr-h*1 and *Br-C*, RT-qPCR was carried out using Applied Biosystems 7500 Fast Real-Time PCR system (ABI, Carlsbad, CA, USA). Each expression level was normalized by *ribosomal protein gene L13a* (*RPL13a*) and *elongation factor 1α* (*EF1-α*) expression [28,29]. The gene-specific primers used for the RT-qPCR were designed by ORF sequence using Primer 5 software, as listed in Appendix A. RT-qPCR thermocycler conditions were 95 °C 3 min; 95 °C 15 s, 58 °C 15 s, 72 °C 15 s, 40 cycle; 95 °C 15 s, 60 °C 1 min, 95 °C 15 s.

### 2.5. dsRNA Preparing

For dsRNA synthesis, the partial cDNA sequence of *RaSsp1* target gene was massively amplified using gene-specific primer, and the amplified product was recycled with SanPrep Column DNA gel Recovery Kit (Sangon Biotech, Shanghai, China), and then the recycled fragment was subcloned into a pMD^®^18-T Vector. The DH5α competent cell (Sangon Biotech, Shanghai, China) was transferred into a recombinant plasmid. The bacteria were evenly spread onto LB solid ampicillin medium and then cultured in an incubator until the plaque grew at 37 °C. Several single colonies were selected for PCR amplification to screen bacterial fluid containing the recombinant plasmid with target fragment, and then they were cultured in an oscillator at 37 °C and 200 rpm for 14 h. Plasmids were extracted using SanPrep Column Plasmid Mini-Preps Kit (Sangon Biotech, Shanghai, China). The target fragment was amplified with primers T7 promoter at the 5′ end using the plasmid as template. PCR products were extracted by phenol chloroform, and dsRNA was synthesized using purified product as a template, which was also purified with phenol chloroform extraction. The purity, concentration, and quality of dsRNA were confirmed in the same way as RNA detection. For the RNAi experiment, EGFP was selected as a control gene, and dsRNA was generated using the EGFP vector pQBI-polII (Wako, Osaka, Japan).

### 2.6. RNAi Experiment

With regard to the fact that gut-purged individuals generally continue 3–4 days to initiate molting, 500 nL dsRNA (2500 ng/μL) (*EGFP* or *RaSsp1*) was intraperitoneally injected into the termites at the beginning of the gut-purging period using a Model PV 820 micro-injector (World Precision Instruments, Sarasota, FL, USA).

To observe the morphological effects, newly molted presoldiers and soldiers were collected every 24 h after dsRNA injection and were maintained in a new petri dish. Meanwhile, 4 times the amount of workers were placed to tend to the presoldiers and soldiers. The presoldiers and soldiers were photographed, and their local traits, including the length and width of the head capsule, the length and width of the pronotum, and the length of left of the mandible were measured under Stereo microscope Leica 205a (Leica Microsystems, Wetzlar, Germany), according to the measurement standard of specific weapons described in previous reports [17,36]. Every treatment was carried out 3 times. The digital photos were overlaid and connected using software Helicon Focus v7.6.6. For gene expression analysis after the RNAi treatment, the individuals were collected 3 days after injection with *RaSsp1* dsRNA.

### 2.7. Statistical Analysis

All values are expressed as the mean ± SD, and the graph was created with GraphPad Prism 9.0.1. One-way analysis of variance (ANOVA) was used to statistically analyze the above data, the significant differences among the multiple treatments were compared with LSD multiple comparison test, and between-paired comparisons were analyzed by Student’s *t* test. All the above data analyses were performed with IBM SPSS Statistics Version 20 at the *p* < 0.05 level.

## 3. Results

### 3.1. Characterization and Bioinformatics Analysis of RaSsp1

*RaSsp1* was screened from the transcriptome database of *R. aculabialis* based on its abnormally high expression in soldiers and was certified by sequence cloning. The ORF of *RaSsp1* comprised 741 bp, encoding 246 amino acids (GenBank accession no. UES72773.1) (Figure 1A). RaSsp1 had the highest homology with AB195158.1 (*Nasutitermes takasagoensis* soldier-specific protein1 (*Ntsp1*)) and XM-022069066.1 (*Zootermopsis nevadensis* takeout-like protein) according to a search of the NCBI database. The predicted molecular weight of RaSsp1 was 27.8 kDa, with the isoelectric point being 6.32. The RaSsp1 N-terminal 20 amino acids were predicted to be a signal peptide with SignalP 4.1. The secondary structure contained four alpha helices (33.3%) and eight beta folds (41.1%) using the online serves (PredictProtein) (Figure 1A,B), as does the JHBP domain-containing protein.

Regarding its three-dimensional structure, RaSsp1 had an elongated β-barrel fold conformation, consisting of a long spinal helix wrapped in a highly curved β-sheet, which was suitable for transporting hydrophobic JH (Figure 1B). JHIII was anchored in the hydrophobic cage as the epoxide of the JHBP-bound JH according to the protein ligand molecular docking online tool AutoDock (Figure 1C). A phylogenetic analysis was performed for RaSsp1 and other similar proteins, including insect takeout and JHBP proteins, by the maximum likelihood method in MEGA7 software (Figure 1D). The phylogenetic relationships of RaSsp1 were phylogenetically closest to Ntsp1 (BAD91203.1) of *Nasutitermes takasagoensis*, followed by takeout (XP_021924758.1) of *Zootermopsis nevadensis*.

### 3.2. Temporospatial Expression Pattern of RaSsp1

RT-qPCR was performed to analyze *RaSsp1* expression in a variety of termite castes and tissues. As shown in Figure 2A, in natural colonies, a notable enhancement in the level of *RaSsp1* expression in the heads of soldiers (240.45 ± 51.77) was observed, compared to that of workers (0.94 ± 0.82) and presoldiers, with the relative expression levels in the head of soldiers nearly 255 times higher than that of workers (*p* < 0.01). Meanwhile, *RaSsp1* was expressed mainly in the abdomen and head, in particular the former (1.48 ± 0.32) (Figure 2B) (*p* < 0.01). Furthermore, *RaSsp1* exhibited tissue specificity in the head and abdomen, as evidenced by the finding that *RaSsp1* was expressed mainly in the mandible, hemolymph, and then in the head cuticle, with its expression level of the mandible being highest (2.08 ± 0.39) and the midgut being low (*p* < 0.01) (Figure 2C).

The following expression of *RaSsp1* was quantified at various developmental stages of soldier differentiation using JHA-induced termites. As shown in Figure 2D, during the differentiation of worker–presoldier–soldier, except for a small peak (0.34 ± 0.01) within 24 h after the 1st molting, *RaSsp1* expression decreased steadily and then began to increase sharply on the day before the 2nd molting. *RaSsp1* expression was 0.69 ± 0.05 on the day before molting, 3.79 ± 0.38 within 24 h after molting, and 27.09 ± 0.78 on the first day after molting, eventually reaching a peak at 129.17 ± 18.74 on the last day, detected after the 2nd molting (*p* < 0.01) (Figure 2D).

### 3.3. Effects of RaSsp1 RNAi on Presoldier Development before Worker Molting

To evaluate the effects of RaSsp1 on presoldier differentiation, *RaSsp1* dsRNA was injected at the beginning of the worker gut-purging period. It could be clearly seen that, after *RaSsp1* RNAi, the *RaSsp1* expression level significantly decreased by 89%, from 1.01 ± 0.08 to 0.11 ± 0.03 (*p* < 0.01) (Figure 3A). The molting ratio of *RaSsp1* RNAi-treated workers decreased from 48.3% to 23.3%, and the molting peak was significantly delayed from the 5th to the 6th day (Figure 3B). *RaSsp1* RNAi shortened the head capsule from 1.26 ± 0.16 to 1.22 ± 0.15 mm (Figure 3C,D) and the mandible from 0.62 ± 0.06 to 0.59 ± 0.05 mm; However, the differences compared to the controls were not significant (Figure 3E) (*p* > 0.05).

In addition, the expression levels for JH signal related genes were detected using RT-qPCR in the head of workers on the 3rd day after injection of RaSsp1 RNA. The expression of *Met* and *Kr-h*1 decreased significantly, compared to the control (*p* < 0.01) (Figure 4A,B). *Met* expression decreased 61%, from 2.70 ± 0.19 to 1.65 ± 0.12. *Br-C* expression was not significantly downregulated (Figure 4C).

### 3.4. Effects of RaSsp1 RNAi on Soldier Development before Presoldier Molting

To further evaluate the role of RaSsp1 in soldier differentiation, *RaSsp1* dsRNA was injected at the beginning of the gut-purging period in presoldiers. On the 3rd day after *RaSsp1* dsRNA injection, *RaSsp1* expression was reduced to 5.64% of the *EGFP* RNAi control group, (*p* < 0.01) (Figure 5A). The molting ratio and duration were affected by *RaSsp1* dsRNA injection. Molting occurred from the 4th to the 6th day after *EGFP* RNAi and was 77.8% complete on the 5th day. However, *RaSsp1* RNAi decreased to 39.2% on the 5th day; 21.6% of termites had not molted by the 9th day (Figure 5B).

In terms of the changes of soldier morphology after dsRNA injection, the head capsule of the *RaSsp1* RNAi soldier became rounder, owing to its length shortening (Figure 5C), as in the presoldier period. *RaSsp1* RNAi significantly reduced the average length of the head capsule of soldiers by 19.6% (from 1.71 ± 0.13 to 1.37 ± 0.12 mm) (*p* < 0.01) compared to the control (Figure 5D) but did not affect its width. Therefore, the length-to-width ratio of the soldier head capsule was significantly decreased by 18.6% (Figure 5E). Meanwhile, *RaSsp1* RNAi significantly shortened the soldier left mandible (from 0.78 ± 0.04 to 0.70 ± 0.06 mm) (Figure 5F) (*p* < 0.01). In addition, the length and width of the soldier pronotum were reduced by 8.6% and 5.9%, respectively, by *RaSsp1* RNAi (*p* < 0.01) (Figure 5G).

Next, we evaluated the effects of *RaSsp1* on the expression of the JH receptor gene *Met*, its key transcription factor *kr-h1*, and the ecdysone signal genes *Br-C* by RT-qPCR. Expression of *Met* decreased on the 3rd day after injection, from 4.02 ± 0.21 to 2.12 ± 0.38 (*p* < 0.01) (Figure 6A). *RaSsp1* RNAi reduced the expression of *kr-h1* by 27% (from 2.66 ± 0.19 to 0.72 ± 0.13) (*p* < 0.01) (Figure 6B) and *Br-C* by 27.35% (*p* < 0.01) (Figure 6C).

## 4. Discussion

In this study, we cloned the sequence of *RaSsp1* and analyzed its structure, expression pattern, function during soldier differentiation in *R. Aculabialis*, and its effects on JH pathway genes. The results indicate that *RaSsp1* was involved in the formation of soldier-specific morphological characteristics (i.e., weapons). This is the first report of the structure and function of *RaSsp1* during termite soldier differentiation.

### 4.1. Characteristic and Functional Prediction of RaSsp1

The division of labor among termite castes maintains the complex society of a eusocial insects. In termite colonies, the soldier caste defends the colony physically by biting or snapping enemies using sclerotized and enlarged mandibles [1]. As a critical endocrine regulator of soldier differentiation, JH requires JHBPs, which can specifically bind and transport JH from the corpora allata to the target tissue, which then induces the formation of specific characteristics of soldiers. Transcriptome analysis of *R. aculabialis* showed a significant difference in *RaSsp1* expression in the heads between worker and soldier castes in *R. aculabialis,* which was the most observed under natural conditions (unpublished); this was confirmed by real-time PCR (Figure 2A). *RaSsp1* is similar to the JHBP domain-containing protein according to the online tool Predict Protein, and it is a homolog of takeout and JHBP-related proteins of other insect species. Takeout proteins have amino acid sequence homology with JHBP, which is a hydrophobic ligand-binding protein in the hemolymph of Lepidoptera that delivers the sesquiterpenoid JH to target tissues [37,38]. Takeout and JHBP have similar α/β barrel and folds, and they show structural conservation of the α1 helix, N-terminal disulfide bond, hairpin, five-stranded twisted β-sheet, and long α3 helix [32,39]. The crystal structure of recombinant *Bombyx mori* JHBP has two JH-binding pockets that bind methyl-2,4-pentanediol 1(MPD1) and MPD2 [39,40]. We found that RaSsp1 had a similar α/β barrel fold and JH-binding pocket as takeout and JHBP. Considered together, RaSsp1 may bind and transport similar long carbon chain molecules in the JH-binding pocket, and then JHIII could be anchored in the same hydrophobic cage (Figure 1C), but the binding region of RaSsp1 and JH III in termites was not the same as JHBP and JH III in *B. mori* [41].

### 4.2. Temporospatial Expression Pattern of RaSsp1 in R. aculabialis

During JHA-induced presoldier differentiation, except for a small peak within 24 h of the 1st molting, *RaSsp1* expression decreased gradually. After JHA treatment, the intrinsic JH titer exceeded that of pseudergates and peaked immediately prior to presoldier molting [11]. Peak *RaSsp1* expression was delayed relative to the peak JH titer the day after the first molting [11]. This later peak of *RaSsp1* may be an independent component of the physiological program leading to presoldier development.

During JHA-induced presoldier–soldier differentiation, *RaSsp1* expression began to increase the day before the 2nd molting and reached 205-fold that in workers on the 9th day after the 2nd molting. Under natural conditions, *RaSsp1* expression is higher in the head of soldiers compared to workers and presoldiers. JHBP is involved in regulating not only insect development but also behavior [42,43,44,45]. JHBP may be closely related to the behavioral flexibility of the queen, such as foraging and caring for eggs, larvae and pupae in *Wasmannia auropunctata* [46], and to the trail-following behavior of workers of *R. flavipes* [47]. However, the JH titer in terminally differentiated soldiers is similar to that in workers and pseudergates [48,49]. Therefore, the upregulated expression of *RaSsp1* before the 2nd molting likely promotes the binding and transport of JH to target tissues. However, its abnormally high expression in soldiers suggests that RaSsp1 has other functions, possibly in the defensive behaviors of soldiers.

Although *Takeout* genes are expressed in brain-associated fat bodies, the antennae and the gut in *Drosophila* [50,51,52], *RaSsp1* was expressed mainly in the mandible, hemolymph, and head cuticle (Figure 2B). *Ntsp1*, the closest homolog of RaSsp1, was expressed in a single layer of secretory cells surrounding the reservoir of the frontal gland [35]. Frontal gland secretion of *Nasutitermes* contains large amounts of terpenoids, which have a basic carbon skeleton and resemble JH structurally [53,54,55]. Differences in anatomical structures may affect the spatial expression of *RaSsp1*, which was highest in the abdomen, and of Ntsp1, which was highest in the head. In *Nasutitermes*, the frontal gland is in the head of the soldier, but in *Reticulitermes*, it extends from the head to the abdomen [56]. Therefore, *RaSsp1* and Ntsp1 likely transport JH in the hemolymph. Regarding the mandible, we speculate that *RaSsp1* is related to the aggressive behaviors of soldiers, including biting, communicating and drumming. Hojo [35] postulated that *Ntsp1* is involved in defense behavior of soldiers and that the *Ntsp1* protein may bind to terpenoids in the front gland. Therefore, *RaSsp1* and terpenoids may cooperatively regulate the defense behavior of soldiers.

### 4.3. The Function of RaSsp1 in Soldier Differentiation

A high JH titer induces soldier differentiation, although different types of JHs or JHAs and termite species affect the efficiency of caste differentiation in termite species [57,58,59]. However, in our present study, the time and rate of molting induced by JH was delayed and reduced, respectively, after injection of *RaSsp*1 dsRNA before the 1st and 2nd molting (Figure 3B and Figure 5B). In addition, JHBP may regulate molting in *B. mori* [60] and larval diapause in *Omphisa fuscidentalis* [61]. RaSsp1 might bind and transport JH via its JH protein binding domain (Figure 1). Therefore, RaSsp1 may promote molting by binding and transporting JH during soldier differentiation.

Although *RaSsp*1 RNAi before the 1st molting (from worker to presoldier) did not affect presoldier morphology, *RaSsp*1 RNAi before the 2nd molting (from presoldier to soldier) induced morphological abnormalities. Head growth was suppressed to yield the typical rounded heads of workers, whereas control soldiers had rectangular heads. Generally, the head capsule of presoldiers becomes elongated and rectangular after the 2nd molting, and the mandible becomes elongated, accommodating major changes in mandible function together with enlargement of the associated mandibular muscles [1,62]. Thus, the effect on head shape of *RaSsp*1 RNAi reflects underlying changes to the mandibular muscles and epidermis. Normal expression of *RaSsp*1 is an important factor not only for soldier differentiation but also for the development of soldier-specific traits during the soldier period. Further morphological and histological analyses of dsRNA-injected individuals are needed to determine the function of RaSsp1 during soldier differentiation.

To explore the effects of *RaSsp*1 RNAi on JH signaling, we analyzed the JH receptor *Met*, the JH-inducible transcription factor *Kr-h1* and the ecdysone -signaling gene *Br-C*. ZnMet RNAi resulted in significantly shorter mandibles and smaller head capsules than in control soldiers [17]. Expression of *Met*, *Kr-h1* and *Br-C* increased abruptly directly after the 1st molting and peaked on the 5th day during the presoldier period [13,17]. In this study, expression of *Met*, *Kr-h1* and *Br-C* were downregulated by *RaSsp1* dsRNA injection, regardless of whether injection was before the first or second molting. In sum, the *RaSsp*1 is involved in the JH-signaling pathway and is crucial for soldier differentiation.

## 5. Conclusions

In this study, *RaSsp*1 was predicted to bind and transport JH by structural analysis, and the functional studies by RNAi suggest that *RaSsp1* played an important role in the formation of soldier-specific traits, including head capsules and mandibles by RNAi for the first time. Knocking down *RaSsp1* affected the expression of JH pathway genes, *Met*, *Kr-h1* and *Br-C*. Thus, a conclusion could be reached that JH was secreted from the corpora allata and then bound to RaSsp1, which transported JH to the hemolymph, eventually regulating soldier differentiation by activating its receptor Met and transcription factor Kr-h1. Our study not only presented new insights in caste differentiation in social insects, it also increased our understanding of the success of this significant species.

## Figures and Tables

**Figure 1 insects-13-00502-f001:**
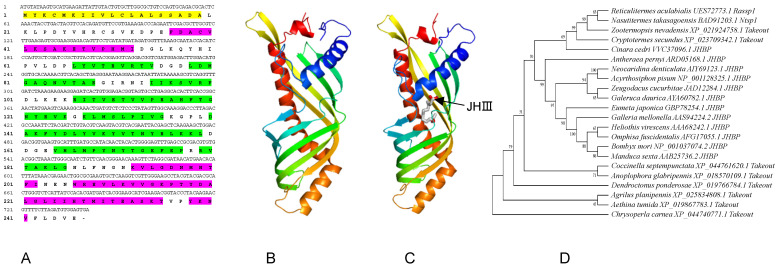
The sequence, predicted topology structures and phylogenetic tree of RaSsP1 in *Reticulitermes aculabialis*. (**A**) The cDNA sequence and the deduced amino acid sequence of *RaSsP1* (GenBank accession no. UES72773.1). Yellow region: signal peptide; purple region: alpha helix; green region: beta fold. (**B**) Predicted 3D structure of RaSsp1. (**C**) Predicted binding model between JHIII and RaSsp1 with AutoDock. JHIII is marked in grey and pointed out with a black arrow. (**D**) The phylogenetic tree of RaSsp1 in insects was constructed using the maximum likelihood method based on amino acid sequences.

**Figure 2 insects-13-00502-f002:**
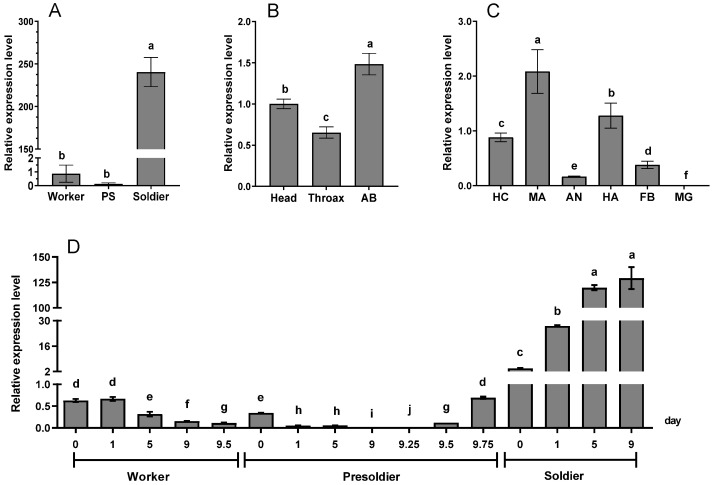
*RaSsp1* expression patterns of *R. aculabialis* in nature colonies and induced soldier differentiation. (**A**) *RaSsp1* expression levels of the head of workers, presoldiers and soldiers in nature colonies. (**B**) Expression levels of *RaSsp1* in soldier different part. (**C**) Expression levels of *RaSsp1* in soldier tissues of nature colonies. PS: presoldier; AB: abdomen; HC: head cuticle; MA: mandible; AN: antennae; HA: hemolymph; FB: fat body; MG: midgut. (**D**) The *RaSsp1* expression patterns in the heads of R. aculabialis during soldier differentiation with JHA induction. Workers: 0, 1, 5, 9, and 9.5 days after JHA treatment; presoldier: 0, 1, 5, 9, 9.25, 9.5 and 9.75 days after the 1st molting; soldier: 0, 1, 5, 9 after the 2nd molting; 0 days in presoldiers and soldiers show within 24 h after molting. Different letters above the bars denote significant differences (one-way ANOVA followed by LSD test using SPSS, *p* < 0.05).

**Figure 3 insects-13-00502-f003:**
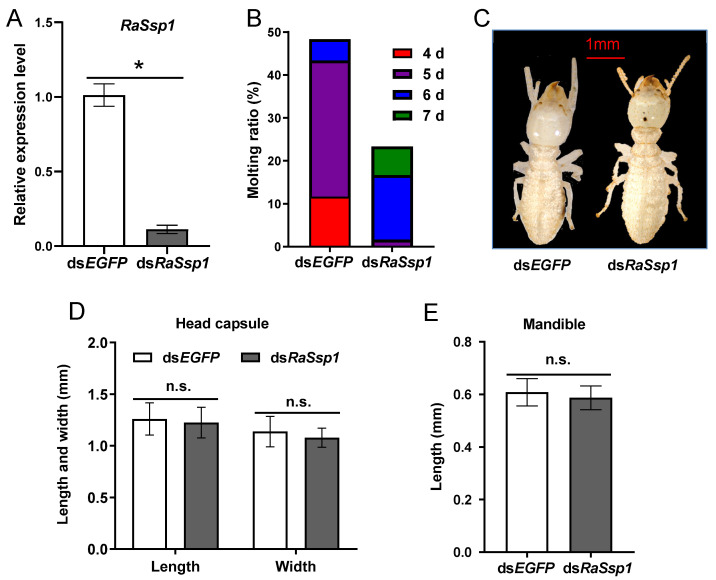
Effect of worker *RaSsp1* RNAi on presoldier differentiation. *RaSsp1* dsRNA was injected at the beginning of the gut-purging period in workers, with *EGFP* dsRNA as control. (**A**) Expression levels of *RaSsp1* in the head of workers on the 3rd day after injection of dsRNA *RaSsp1* (mean ± S.D., technical triplicates). (**B**) Effect of the 1st molting ratio after *Rassp1* RNAi; 4, 5, 6 and 7 d are the 4th, 5th, 6th and 7th days after injection dsRNA (n = 60). (**C**) Presoldier morphology affected by dsRNA injection. Both presoldiers were photographed on the 2nd day after the 1st molt. Scale bar indicates 1 mm. The photos represent the 16 presoldiers. (**D**) The length and width of the head capsule (mean ± S.D., n = 16). (**E**) The left mandible length. Asterisks indicate a significant difference; n.s. means not significant (one-way ANOVA followed by LSD test using SPSS, * *p* < 0.05).

**Figure 4 insects-13-00502-f004:**
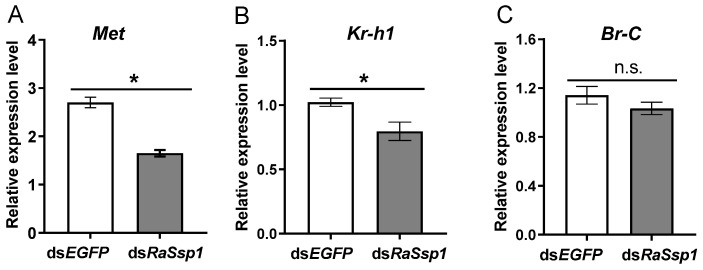
The expression levels of JH signal related gene in the head of workers on the 3rd day after injection of *RaSsp1* dsRNA. *RaSsp1* dsRNA was injected at the beginning of the gut-purging period in workers, with *EGFP* dsRNA as control. (**A**) *Met*, (**B**) *Kr-h1*, (**C**) *Br-C*. Asterisks indicate a significant difference (one-way ANOVA followed by LSD test using SPSS, * *p* < 0.05).

**Figure 5 insects-13-00502-f005:**
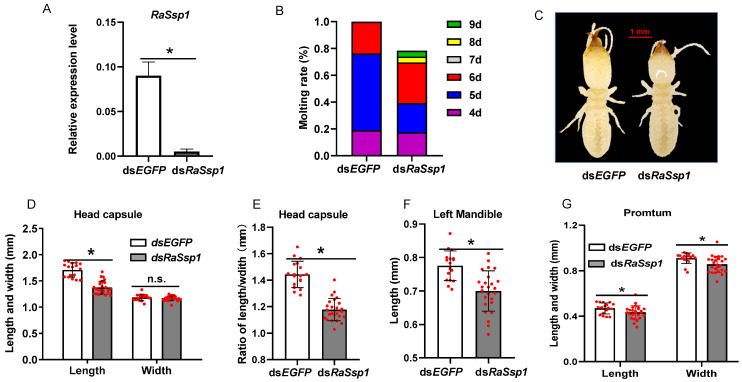
Effects of presoldier *RaSsp1* RNAi on soldier differentiation. *RaSsp1* dsRNA was injected at the beginning of the gut-purging period in presoldiers, with *EGFP* dsRNA as control. (**A**) Expression levels of *RaSsp1* in the head of presoldiers on the 3rd day after injection of *RaSsp1* dsRNA (mean ± S.D., technical triplicates). (**B**) Effect of the 2nd molting ratio after *Rassp1* RNAi; 4, 5, 6, 7, 8 and 9 d are the 4th, 5th, 6th, 7th, 8th and 9th days after injection with dsRNA (n = 46). (**C**) Soldier morphology affected by dsRNA injection. Both soldiers were photographed on the 2nd days after the 2nd molt. Scale bar indicates 1 mm. The photos represent the 36 presoldiers. (**D**) The length and width of the head capsule (mean ± S.D., n = 36). (**E**) The ratio of length/width of the head capsule. (**F**) The left mandible length. (**G**) The length and width of the pronotum. Asterisks indicate a significant difference (one-way or two-way ANOVA followed by LSD test using SPSS, * *p* < 0.05). White and grey columns indicate Rassp1 and EGFP dsRNA-injected individuals, respectively.

**Figure 6 insects-13-00502-f006:**
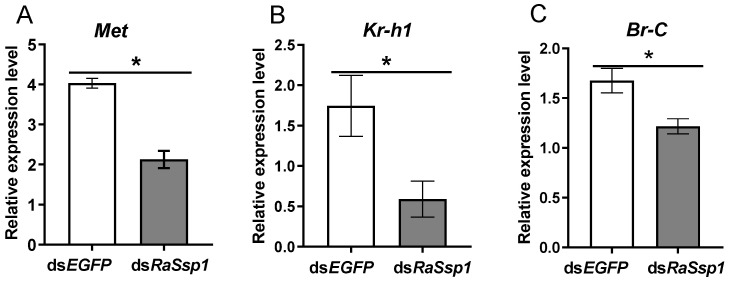
The expression levels of JH signal related gene in the head of presoldiers on the 3rd day after injection of *RaSsp1* dsRNA. *RaSsp1* dsRNA was injected at the beginning of the gut-purging period in presoldiers, with *EGFP* dsRNA as control. (**A**) *Met*, (**B**) *Kr-h1*, (**C**) *Br-C*. Asterisks indicate a significant difference (one-way ANOVA followed by LSD test using SPSS, * *p* < 0.05).

## Data Availability

Anyone can apply for the original data from the corresponding author within reason.

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
