# Peer review of "Soldier Caste-Specific Protein 1 Is Involved in Soldier Differentiation in Termite Reticulitermes aculabialis"

_insects, 2022, doi:10.3390/insects13060502_

Round 1
Reviewer 1 Report
The manuscript by Wu et al. entitled “Soldier caste-specific protein 1 involved in the soldier differentiation in termite Reticulitermes aculabialis” represents a novel observation linking the worker-to-soldier transformation with the function of a gene soldier caste-specific protein 1 (RaSsp1). While the role of major endocrine hormone, i.e. the juvenile hormone (JH), and its signaling pathway is well-studied in termites, the genes associated with the regulation of JH homeostasis are largely unknown. The authors focused on the gene Ssp1 with JH affinity, performed the characterization analysis, the expression analysis using qPCR and function analyses using RNAi. The results show a clear link between RaSsp1 and JH signaling pathway (JH receptor gene and its key transcription factor) and its effect on a soldier-specific morphology. Overall, I consider the manuscript as suitable for this journal, provided that the authors address my comments below.
General comments:
[1] My concern is related to the statistical analyses of the gene expression analysis. The authors have performed "own-way ANOVA" in Fig3 and "two-way ANOVA" in Fig5, but I do think that it is inconsistent methods. In addition, the authors should provide the two explanatory variable and a response variable in two-way ANOVA.
[2] I largely agree with your discussion on the possible role for growth of soldier-specific traits. I am sure, however, the authors did not obtain the results that RaSsp1 expression was the essential factor for the soldier determination. I found the discussion to be a bit speculative.
Minor comments:
L123-124
How many colonies did you use?
L133-134
Please specify a reason that the qPCR was performed using the same cDNA sample (i.e. technical reprication), not different cDNA samples (i.e. biological replication). My view (and that of Vaux et al. 2012 EMBO Rep) is that especially biological replicates are needed.
L157-158
The authors performed additional gene expression analyses with other qPCR machines. Why? I think it is extra work.
L160
Isn't it transcriptome data? (as shown above at line 144)
L183
Why could the authors confirm the gut-purged presoldier? I suppose the presoldier have a white-colored abdomen throughout the developmental stage. Please explain how to obtain the gut-purged presoldier used for dsRNA injection. The JHA induction? The field collection?
L184
How many colonies did you use? I believe that worker individuals collected in a single termite colony are only pseudoreplicates because their genetic background are closely related and share the identical environment.
L258
It would be helpful to all readers to describe supplemental information on JHA induction, i.e., mortality, rate of presoldier/soldier differentiation.
Author Response
May 21, 2022
Pro. Review
Insects
Dear Dr. Liu,
We are grateful for the positive reviews of our paper (Insects-1736552) and the thoughtful, constructive comments from you and the two reviewers. In addition, the language of our article has been edited by Textcheck (https://www.textcheck.com/login) (you can find in the file of “Revised manuscript with Track Changes”) and revised our manuscript to meet the Insects’s style requirements. We are submitting a suitably revised version of the paper. The detailed point-by-point responses to the comments from the reviewers are listed below:
Response to Reviewer 1
Comments and Suggestions for Authors
The manuscript by Wu et al. entitled “Soldier caste-specific protein 1 involved in the soldier differentiation in termite Reticulitermes aculabialis” represents a novel observation linking the worker-to-soldier transformation with the function of a gene soldier caste-specific protein 1 (RaSsp1). While the role of major endocrine hormone, i.e. the juvenile hormone (JH), and its signaling pathway is well-studied in termites, the genes associated with the regulation of JH homeostasis are largely unknown. The authors focused on the gene Ssp1 with JH affinity, performed the characterization analysis, the expression analysis using qPCR and function analyses using RNAi. The results show a clear link between RaSsp1 and JH signaling pathway (JH receptor gene and its key transcription factor) and its effect on a soldier-specific morphology. Overall, I consider the manuscript as suitable for this journal, provided that the authors address my comments below.
Response: We appreciate the positive feedback.
General comments:
[1] My concern is related to the statistical analyses of the gene expression analysis. The authors have performed "own-way ANOVA" in Fig3 and "two-way ANOVA" in Fig5, but I do think that it is inconsistent methods. In addition, the authors should provide the two explanatory variable and a response variable in two-way ANOVA.
Response: Thanks for your good question. One way ANOVA was used for the statistical analyses, we had revised it (Lines 199, 254, 283, 295, 313, 334).
[2] I largely agree with your discussion on the possible role for growth of soldier-specific traits. I am sure, however, the authors did not obtain the results that RaSsp1 expression was the essential factor for the soldier determination. I found the discussion to be a bit speculative.
Response: Thanks for your good question. We also believe it is not convinced that "RaSsp1 expression was the essential factor for the soldier determination", but we can believe that "RaSsp1 expression was the important factor for the soldier determination" (Lines 413).
Minor comments:
L123-124
How many colonies did you use?
Response: Revised as suggested. We use three nature colonies (Lines 125).
L133-134
Please specify a reason that the qPCR was performed using the same cDNA sample (i.e. technical reprication), not different cDNA samples (i.e. biological replication). My view (and that of Vaux et al. 2012 EMBO Rep) is that especially biological replicates are needed.
Response: Revised as suggested. In this experiment, Each treatment was repeated 3 times (biological replication, 5 different individuals/per replication), and three RT-qPCR assays (technical replication) were performed using the same cDNA samples of each time point. (Lines 137-139).
L157-158
The authors performed additional gene expression analyses with other qPCR machines. Why? I think it is extra work.
Response: Revised as suggested (Lines 144). The extra qPCR machine has been deleted.
L160
Isn't it transcriptome data? (as shown above at line 144)
Response: It is ORF sequence, not transcriptome data. It has been revised (Lines 161).
L183
Why could the authors confirm the gut-purged presoldier? I suppose the presoldier have a white-colored abdomen throughout the developmental stage. Please explain how to obtain the gut-purged presoldier used for dsRNA injection. The JHA induction? The field collection?
Response: During soldier differentiation, termites will undergo a gut-purged period about 3 days before per molting, in which termite exclude intestinal contents (the color of abdomen change from brown-to-white) and no longer feed. So, in order to observed clearly the gut-purged period, we added brilliant blue solution to filter paper, the abdomen of termite in gut-purged period will change color from blue to white (lines 121-124). The termite was induced with JHA induction.
L184
How many colonies did you use? I believe that worker individuals collected in a single termite colony are only pseudoreplicates because their genetic background are closely related and share the identical environment.
Response: We use one colony in dsRNA experiment. You were dead right. We collected worker from one colony for making sure the genetic identity (lines 183-185).
L258
It would be helpful to all readers to describe supplemental information on JHA induction, i.e., mortality, rate of presoldier/soldier differentiation.
Response: In the preliminary experiment, we had published the data of mortality, rate of presoldier/soldier differentiation on JHA induction in Reticulitermes aculabialis (Chu, J.; Wu, Z.; Du, Y.; Xi, Y.; An, S.; Su, L. Characteristics of soldier differentiation of termite Reticulitermes aculabialis induced by juvenile hormone and expression profiling of juvenile hormone-related genes. J. Plant Prot. 2020, 47, 1099–1107.). (lines 470-471)
Best regards,
Lijuan Su, Xinming Yin
Henan Agricultural University, Zhengzhou, China
Email: xinmingyin@hotmail.com (X.Y.); sulijuan816@126.com (L.S.)

Reviewer 2 Report
I am confident that the article “Soldier caste-specific protein 1 involved in …” communicates high value scientific information and that all experiments were carefully executed and analyzed. The conclusions are justified, and the results are presented in acceptable formats However, this only applies for the graphics.
I am not a native English-speaking person myself, but I found the English too insufficient, which prohibited following the content of the manuscript in multiple parts. That is why I marked “quality of presentation” as “low”, and that the presentation of the results must be improved.
Again, the content of the article is of high value, but the language needs much improvement to give the article the ranking it deserves.Author Response
May 21, 2022
Pro. Review
Insects
Dear Dr. Liu,
Thank you very much for your feedback. We are grateful for the positive reviews of our paper (Insects-1736552) and the thoughtful, constructive comments from you. In addition, the language of our article has been edited by Textcheck (https://www.textcheck.com/login) (you can find in the file of “Revised manuscript with Track Changes”) and revised our manuscript to meet the Insects’s style requirements. We are submitting a suitably revised version of the paper. The detailed point-by-point responses to the comments are listed below:
Response to Reviewer 2
Comments and Suggestions for Authors
I am confident that the article “Soldier caste-specific protein 1 involved in …” communicates high value scientific information and that all experiments were carefully executed and analyzed. The conclusions are justified, and the results are presented in acceptable formats. However, this only applies for the graphics.
Response: We appreciate the positive feedback.
I am not a native English-speaking person myself, but I found the English too insufficient, which prohibited following the content of the manuscript in multiple parts. That is why I marked “quality of presentation” as “low”, and that the presentation of the results must be improved.
Response: Thanks for your good suggestions. The language of our article has been edited by Textcheck (https://www.textcheck.com/login).
Again, the content of the article is of high value, but the language needs much improvement to give the article the ranking it deserves.
Response: We appreciate the positive feedback.
Best regards,
Lijuan Su, Xinming Yin
Henan Agricultural University, Zhengzhou, China
Email: xinmingyin@hotmail.com (X.Y.); sulijuan816@126.com (L.S.)
